

# The DUF348 domains of resuscitation promoting factor 2 play important roles in the enzymatic and biological activities in *Rhodococcus erythropolis* KB1

Jianhui Fu[1], Jixiang Chen[1], Yonggang Wang[2], Dan Luo[1], Tianfeng Wang[1] and Qingfang Zhang[1]

[1] School of Petrochemical Engineering, Lanzhou University of Technology, Lanzhou, Gansu province, China
[2] School of Life Science and Engineering, Lanzhou University of Technology, Lanzhou, Gansu province, China

Corresponding authors
Jixiang Chen, betcen@163.com
Yonggang Wang, wangyg@lut.edu.cn

## ABSTRACT

*Rhodococcus erythropolis* KB1 is a member of the Actinomycetota and a petroleum-degrading bacterium, isolated from soil contaminated with petroleum products. The resuscitation-promoting factors (Rpf) widely exist among Actinomycetota, which revive the viable but nonculturable (VBNC) state cells and facilitate growth of normal cells. The Rpf2 of the *R. erythropolis* KB1 is the most complex Rpf protein, which consists of the conserved Rpf domain, one G5 domain and three DUF348 domains. The protein demonstrates muralytic activity and growth-promoting and resuscitation effect, but the exact roles of these DUF348 domains in the enzymic and biological activities remain unclear. In this paper, the recombinant plasmids containing *rpf* 2 genes with different DUF348 domain deletion were constructed and expressed in *Escherichia coli*. The enzymatic and biological activities of the mutated Rpf2 proteins were examined. The results showed that the enzymatic activities of the mutated Rpf2 proteins with 1, 2, and 3 DUF348 deletion decreased by 26.27%, 38.17%, and 42.56% respectively when compared with that of the wild-type Rpf2. A negative correlation between the number of DUF348 deletions and the growth-promoting and resuscitation effect on *R. erythropolis* KB1 cells were also observed. The muralytic activities of the mutated Rpf2 proteins showed stability at the temperature range of 20 °C to 40 °C, but showed sharp declines at 50 °C, with the activity dropping by 50.07% to 90.06%, and complete loss at 70 °C and 80 °C, underscoring importance of the DUF348 in thermal stability of the Rpf2. $Zn^{2+}$ and $Mn^{2+}$ slightly enhanced the muralytic activity, while $Mg^{2+}$, $Ca^{2+}$ and $Co^{2+}$ had negligible effects. These findings offered significant insights into mechanism of the Rpf action, emphasizing the critical role of the DUF348 domain.

## INTRODUCTION

When faced with unfavorable environmental conditions, bacteria transition from an active, replicating state to a dormant, non-replicating state. This state is commonly known as the viable but non-culturable state (VBNC) (*Xu et al., 1982*; *Xie et al., 2021*). The

VBNC bacteria exhibit reduced metabolic activity compared to normal cells and typically possess altered cell wall structures, often thicker cell walls (*Vermassen et al., 2019*). The VBNC bacteria can recover from the dormancy state to resume their active growth and reproduction when exposed to favorable conditions (*Zhang et al., 2021*).

The main component of the cell wall is peptidoglycan. Peptidoglycan is composed of alternating residues of N-acetyl muramic acid (MurNAc) and N-acetyl glucosamine (GlcNAc), forming a sugar chain framework, as well as short peptides extending from the lactate group of MurNAc residues (*Bodor et al., 2020*). The peptide stems from different sugar chains that can be directly linked or connected by varying lengths of amino acid linkers. These peptide bridges cross-link parallel chains, creating a sturdy structure that maintains the integrity of the cell membrane (*Bouhss et al., 2008*). Peptidoglycan synthesis is relatively inactive in VBNC state cells. However, in active cells, peptidoglycan exhibits highly dynamic characteristics. Cell wall lysis is an important process for cell growth and is required for the insertion of new peptidoglycan (*Fisher & Mobashery, 2020*).

For many dormant cells such as *Streptomyces coelicolor*, *R. erythropolis*, resumption of active growth requires destruction of their thick protective cell walls. Different bacteria have evolved distinct strategies to accomplish this (*Yan et al., 2021*). *R. erythropolis* KB1 is a member of the Actinomycetota and a petroleum-degrading bacterium isolated from petroleum contaminated soil. The *R. erythropolis* cells in a VBNC state can transition back to a normal state with aid of a protein known as the Rpf (*Fu et al., 2022*). The Rpf was originally discovered in *Micrococcus luteus*, and it possesses the ability to revive VBNC state cells and facilitate the growth of cells in the normal state (*Mukamolova et al., 1998*). The Rpf proteins are widely present among Actinomycetota (*Han et al., 2023*), Some Actinomycetota encoding multiple Rpf proteins. These Rpfs collectively contribute to the reawakening of VBNC state bacteria. In addition to the conserved Rpf domain, various Rpf proteins consist of a range of structural domains, including LysM, LytM, DUF348, G5 (*Sexton et al., 2015*). The specific functions of certain auxiliary domains within Rpf proteins are still not fully understood at this juncture (*Sexton et al., 2020*). It is imperative to achieve a complete characterization of the Rpf itself and systematically assess the contributions of different structural domains.

In this study, we constructed expression plasmids containing *rpf2* genes with varying numbers of DUF348 domain deletions. The recombinant proteins were expressed and their enzymatic and biological activities were investigated. Our aim was to explore the key role of the DUF348 auxiliary domain of Rpf2 in its enzyme activity and biological activities.

## MATERIALS AND METHODS

### Bioinformatics analysis of the different Rpf proteins

The AlphaFold 3 was used to predict the tertiary structure of Rpf2 and its mutant proteins of *R. erythropolis* KB1 (*Abramson et al., 2024*). The protein tertiary structure visualization was employed by VMD software (*Humphrey, Dalke & Schulten, 1996*). The Interpro was used for domain prediction and DOG was used for visualization (*Ren et al., 2009*; *Paysan-Lafosse et al., 2023*).

## Bacterial strains and growth conditions

*R. erythropolis* KB1 (NCBI accession number: CP050124.1) was isolated from petroleum-contaminated soil and preserved in environmental biotechnology laboratory, school of petrochemical engineering, Lanzhou University of Technology. The bacterial cells were cultivated on Lysogeny-broth (LB) solid medium at 30 °C. The *E. coli* strains were also grown on a solid LB medium at the same temperature. The bacterial growth was monitored at $OD_{600}$ nm using a UV-visible spectrophotometer (Unico UV-5120) at predefined time intervals.

## Construction of the expressing plasmids containing the *rpf* 2 genes with different domain deletions and their expression

We first explored the sequence characteristics of Rpf2 and its variants lacking 1, 2, and 3 DUF348 domains. These sequences of the wild-type Rpf2 and its variants were then synthesized by Nanjing Zhongding Biotechnology Co., Ltd., and the expressing pET28a (+) plasmids containing *rpf2* genes with different DUF domain delusions were constructed. The constructed expression vectors were verified by restriction enzyme cleavage and sequencing. The verified plasmids were successfully transformed into *E. coli* BL21 (DE3) (*Studier & Moffatt, 1986*).

*E. coli* BL21 (DE3) cells harboring the pET-28a (+)-*rpfs* plasmid were inoculated into five mL of LB liquid medium supplemented with 50 μg/mL kanamycin and cultured at 37 °C with shaking overnight. This culture was then transferred to 500 mL of the same medium (containing 50 μg/mL kanamycin). Induction was carried out with 0.6 mmol/L isopropyl-$\beta$-D-thiogalactoside (IPTG) at 20 °C for 5.5 h. Subsequently, the Rpf protein was purified using a $Ni^{2+}$ agarose affinity chromatography column and analyzed by SDS-PAGE with Coomassie brilliant blue staining.

## Enzymatic and biological activity assays of the recombinant Rpf proteins

The muralytic activities of the recombinant Rpf proteins were evaluated using the fluorescent substrate 4-methylumbelliferyl- $\beta$-D-N, N′, N″-triacetyl chitotrioside (Sigma, Germany). The enzymatic activities of the Rpfs were quantified by measuring the production of 4-methylumbelliferone from substrate hydrolysis in a 96-well microplate. In each well, 100 μL of substrate buffer containing the substrate was combined with 5 μL of the purified recombinant Rpf proteins. The reaction mixture was incubated at 37 °C in a water bath for 30 min. The reaction was stopped by adding 100 μL of glycine-sodium hydroxide termination buffer. The fluorescence intensity of the resultant 4-methylumbelliferone was measured using a microplate reader with an excitation wavelength of 355 nm and emission wavelength of 460 nm. The measured values were compared against a standard curve of 4-methylumbelliferone. One unit (U) of the enzyme activity was defined as the amount of enzyme producing 1.0 nmol of product per microgram of the fusion protein per hour under these conditions (*Li et al., 2017*).

## Effect of temperature and metal ions on the muralytic activities of the purified Rpf proteins

To evaluate the effect of temperature on the stability of enzyme activity, The Rpf proteins were incubated at different temperatures of 20 °C, 30 °C, 40 °C, 50 °C, 60 °C, 70 °C, and 80 °C for 30 min, followed by measurement of their muralytic activity.

Individual stock solutions of zinc chloride ($ZnCl_2$), magnesium chloride ($MgCl_2$), cobalt chloride ($CoCl_2$), calcium chloride ($CaCl_2$), and manganese chloride ($MnCl_2$) were prepared at a concentration of 10 mmol/L. Subsequently, each metal ion solution was introduced into the reaction system, with a final metal ion concentration of 0.1 mmol/L. The reaction mixtures were then incubated at 37 °C for 30 min. A control sample, containing the Rpf protein reaction system without metal ions was also prepared to establish a baseline for comparison. The muralytic activity of the Rpf protein in each sample was quantified as previously described method.

## The growth-promotion effect of the recombinant Rpf proteins on *R. erythropolis*

To investigate the promoting effect of the recombinant Rpf proteins on *R. erythropolis* cells, 5 μL of logarithmic-phase *R. erythropolis* KB1 cells was added to five mL of LB liquid medium. The recombinant Rpf proteins were then added at a concentration of 10 nM, all the proteins were sterilized through a 0.22 μm filter. Inactivated recombinant Rpf protein was used as the control group (the protein was inactivated in boiling water for 5 min). The cultures were incubated in a shaking incubator at a constant shaking rate of 180 rpm and temperature of 30 °C. At specified time intervals (0, 6, 12, 24, 36, 48, 60, 72, and 84 h), the samples of 0.2 mL were withdrew and optical density of the cultures was measured at 600 nm using a UV-visible spectrophotometer. All the experiments were run in three replications.

## Effect of the recombinant Rpf proteins on resuscitation of the VBNC *R. erythropolis* cells

The method for prepare VBNC cells was described as follow: The *R. erythropolis* KB1 cells were inoculated into five mL of LB liquid medium and cultured overnight. The culture was centrifuged at 3,500 × g for 5 min, and the resulting pellet was resuspended in physiological saline. The washing procedure was repeated for three times. The washed cells were then inoculated into 500 mL of sterile physiological saline and incubated at 4 °C to induce the VBNC state. The VBNC cells of *R. erythropolis* KB1 used in this study had maintained at 4 °C for approximately 2,000 days (*Luo et al., 2019*). In each experiment, five mL of the VBNC *R. erythropolis* KB1 cells was transferred into a sterile tube. A solution of the recombinant Rpf protein filtered through a 0.22 μm membrane was then added at a concentration of 10 nM, followed by a sterilized yeast extract solution (5 g/L) at a final concentration of 0.025% (v/v). The inoculated tubes were incubated at 180 rpm and 30 °C. Samples of 0.1 mL were withdrew at intervals of 0, 12, 24, 36, 48, 60, 72, and 84 h and measured for optical density with a UV-visible spectrophotometer at 600 nm. The culturable cells were also determined with plate counting method. The inactivated recombinant Rpf2 was used as a negative control.

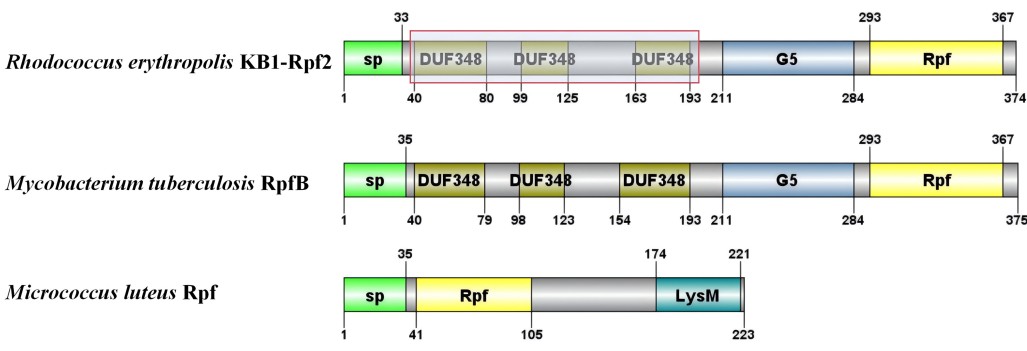

**Figure 1** Schematic diagram of the partial structural domains of Rpf proteins. The red frame shows the deletion of the DUF348 structural domain for this study.

## Statistical analysis

All experiments were performed in three replications and the results were presented as the mean ± standard deviation (SD). One-way ANOVA and comparison of means with Tukey's honestly significant difference post-hoc test ($P < 0.05$) was used to analyze the experimental results (Graphpad Prism, La Jolla, CA, USA). The graphs were constructed with Origin 2018 software (OriginLab Corp., Northampton, MA, USA) and Graphpad Prism.

## RESULTS

### Bioinformatic analysis of the different Rpf proteins of *R. erythropolis* KB1

The critical domains and tertiary structure of the Rpf2 protein from *R. erythropolis* KB1 and its variants lacking the DUF348 domain were analyzed (pTM scores from Alphafold3 are shown in Table S2). In the wild-type Rpf2, the catalytic domain (Rpf domain), three tandem DUF348 domains, and the G5 structure were observed (Fig. 1). In the absence of the DUF348 domains, the Rpf domain remained unaltered, whereas the G5 structure exhibited increased flexibility, and the N-terminal conformation was altered. The loss of the DUF348 domains may weaken their direct or indirect interaction with the G5 structure, which could impair overall functional coordination and lead to a reduction in enzymatic activity (Fig. 2). This evidence suggests that the DUF348 domains play a critical role in maintaining protein structure and stability. Furthermore, a comparative structural analysis was performed on Rpf2 from *R. erythropolis* KB1 and RpfB from *Mycobacterium tuberculosis*. This structural similarity suggests the conservation of functional domains, potentially indicating conserved roles in bacterial physiology across species.

### Expression and enzyme activities of the recombinant Rpf proteins

We successfully constructed, expressed, and purified the full-length Rpf2 protein as well as the different DUF348 domain deleted proteins. The SDS-PAGE gel electrophoresis results of the purified recombinant proteins were show in Fig. 3, and the protein bands in the gel highly matched our expectations.

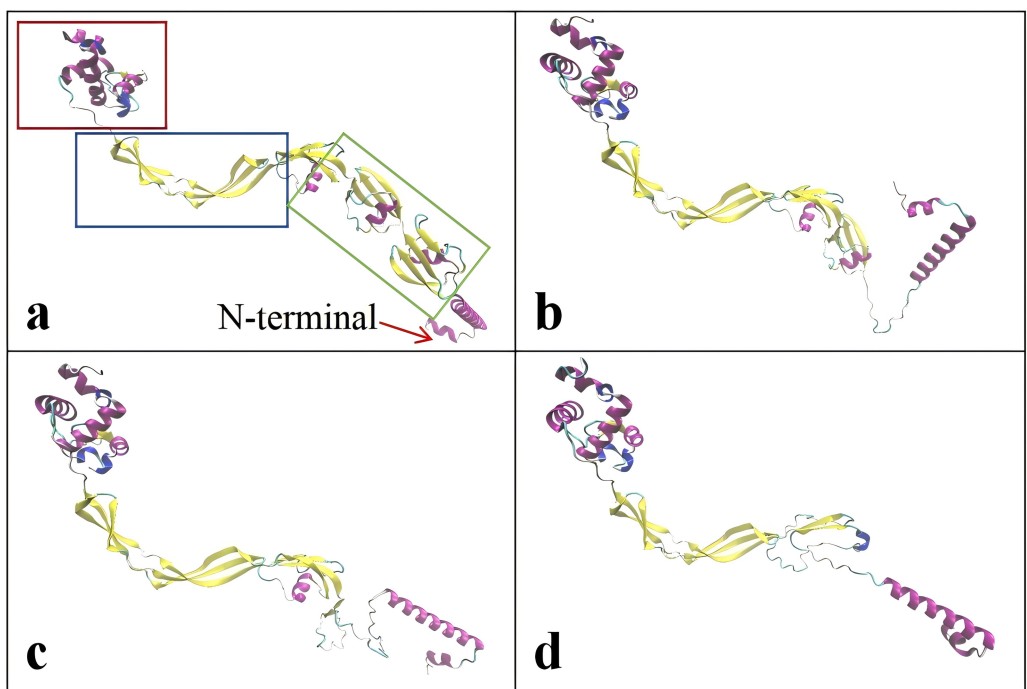

**Figure 2** **Tertiary structure simulation of the different Rpf2 proteins in *R. erythropolis* KB1.** (A) Wild-type Rpf2; (B) Rpf2 deleting 1 DUF348 domain; (C) Rpf2 deleting 2 DUF348 domains; (D) Rpf2 deleting 3 DUF348 domains. The Rpf domain is marked by red frame, the G5 domain is marked by blue frame, the three DUF348 domains are marked by green frame, and a red arrow is added to indicate the N-terminal.

The standard muralytic activity assay method was used to evaluate the catalytic function of the recombinant proteins. All of the purified Rpf proteins showed significant muralytic activities (Fig. 4), but an inverse relationship between muralytic activity and the number of deleted DUF348 domains was observed, indicating the pivotal roles of the DUF348 domains in catalytic activity of the Rpf protein.

## Muralytic activity of the recombinant Rpf proteins at different temperatures and metal ions

The muralytic activities of the recombinant Rpf2 protein and its three variants (1 ΔDUF348, 2 ΔDUF348, and 3 ΔDUF348) were evaluated at a temperature range of 20 °C to 80 °C (Fig. 5). The optimal temperature for maximal muralytic activity of the Rpf2 proteins was observed at 30 °C, the activity of native Rpf2 reached its peak activity of 755 U, while the activities of the different Rpf2 variants were lower than that of the Rpf2. The activities of all proteins decreased with the increasing of temperature. The native Rpf2 consistently exhibited higher muralytic activity than its variants at each temperature point. As the temperature increased to 40 °C and 50 °C, all proteins displayed a significant decline in activities, with native Rpf2 retaining relatively higher activity levels (approximately 724 U and 377 U, respectively) compared to its variants, which indicated the important role of the DUF348 domains in the structural and functional stability of Rpf2, especially under optimal and elevated temperatures.

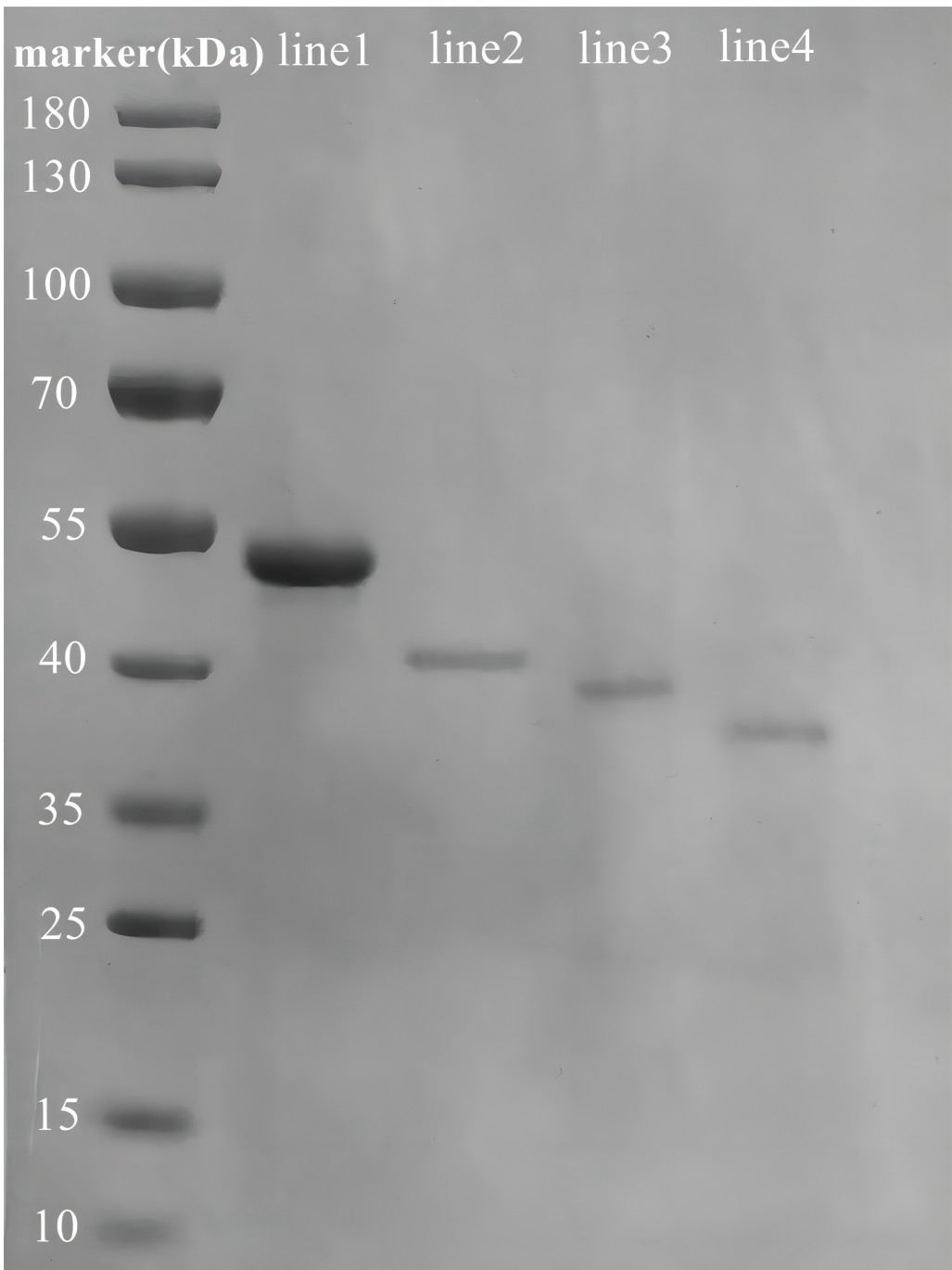

**Figure 3** **SDS-PAGE analysis of the purified recombinant Rpf2 proteins.** Line 1 represents wild-type Rpf2; line 2 represents Rpf2 deleting 1 DUF348 domain; line 3 represents Rpf2 deleting 2 DUF348 domains; line 4 represents Rpf2 deleting 3 DUF348 domains.

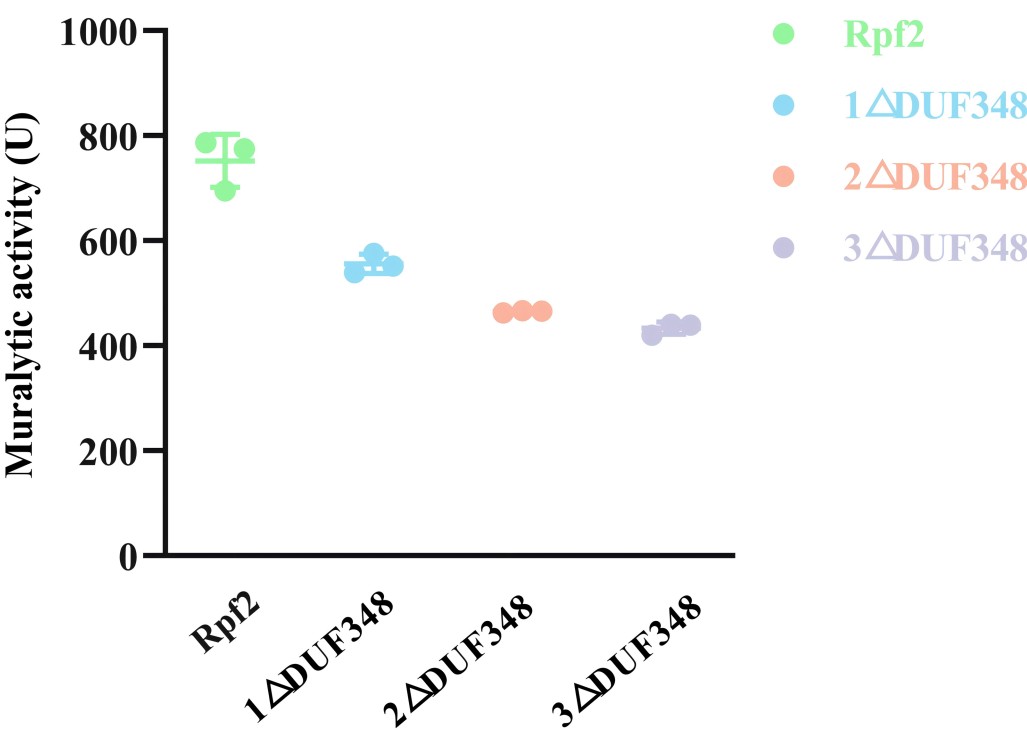

**Figure 4** **Muralytic activities of the different Rpf2 proteins.** 1 △DUF348 represents Rpf2 with one missing DUF348 domain; 2 △ DUF348 represents Rpf2 with two missing DUF348 domains; 3 △ DUF348 represents Rpf2 with three missing DUF348 domains. A total of 1 U of enzyme activity is defined as the amount of enzyme that produces 1.0 nmol of product per microgram of fusion protein per hour.

We also assessed the effect of different metal ions on the muralytic activity of the recombinant Rpf proteins. We found that addition of $Zn^{2+}$ and $Mn^{2+}$ led to a statistically significant enhancement of the muralytic activity of Rpf proteins ($p < 0.05$), indicating that these ions play a crucial role as potent cofactors. The enhancement of muralytic activity by $Zn^{2+}$ and $Mn^{2+}$ was less pronounced in the variants compared to the native Rpf protein, suggesting that the DUF348 domain may play a role in optimizing the binding and interaction of these metal ions with the enzyme. $Mg^{2+}$, $Co^{2+}$, and $Ca^{2+}$ showed little increases in enzymic activities ($p > 0.05$) (Fig. 6).

## Effects of the recombinant Rpf proteins on the growth of *R. erythropolis* KB1

We evaluated the effects of the recombinant proteins on the growth of *R. erythropolis* KB1 cells. It was found that the *R. erythropolis* populations in the groups without Rpf2 or containing the inactivated Rpf2 kept the same. In contrast, addition of the recombinant Rpf2 protein and its mutants significantly promoted growth of the *R. erythropolis* KB1 cells. Notably, the growth promotion effect was positively correlated with the number of DUF348 domains in these proteins (Fig. 7). The wild-type Rpf2 and mutants lacking different amounts of DUF348 had significant differences in their growth-promoting effects on *R. erythropolis* KB1 (Table S3). The intact Rpf2 protein has the most pronounced

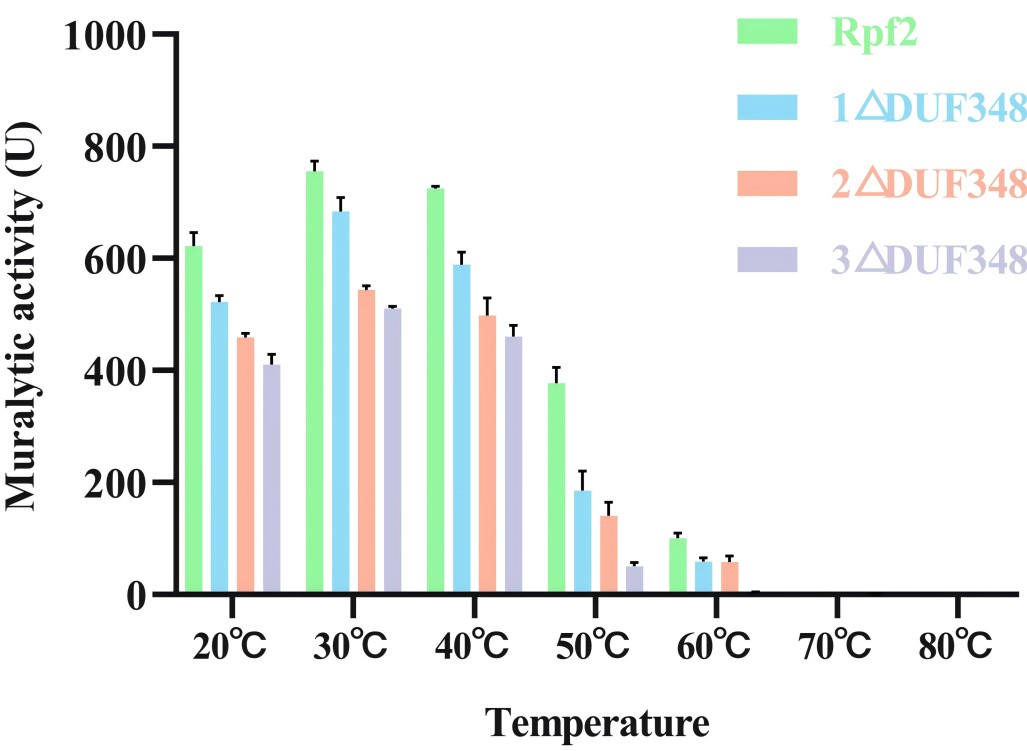

**Figure 5** **Effects of temperatures on the muralytic activities of the different Rpf2 proteins.** 1 △DUF348 Rpf2 represents Rpf2 with one missing DUF348 domain; 2 △DUF348 Rpf2 represents Rpf2 with two missing DUF348 domains; 3 △DUF348 Rpf2 represents Rpf2 with three missing DUF348 domains. A total of 1 U of enzyme activity is defined as the amount of enzyme that produces 1.0 nmol of product per microgram of fusion protein per hour.

growth-promoting effect at the same molar amount, this effect is diminished as the number of DUF348 structural domains decreases. This observation is consistent with our muralytic activity assay and previous experiments, confirming a positive correlation between the muralytic activities of the Rpf proteins and their abilities to promote cell growth (*Luo et al., 2019*).

## Resuscitation effect of the recombinant Rpf proteins on the VBNC state cells of *R. erythropolis* KB1

Resuscitation effect of the different recombinant Rpf proteins on the VBNC state cells was observed by addition of the recombinant Rpf proteins in the VBNC *R. erythropolis* KB1 cells. Our results showed that the growth patterns of the experimental groups (without recombinant Rpf protein) and the control group (with inactivated protein) were similar, neither demonstrated a significant resuscitation effect on VBNC *R. erythropolis* KB1 cells (Fig. 8). The experimental groups with the recombinant Rpf proteins showed pronounced resuscitation and growth promotion effects on the VBNC cells. Further analysis indicated an inverse relationship between the number of missing DUF348 domains and the resuscitation level, highlighting the importance of DUF348 domains in Rpf protein functionality. Deletion of these domains significantly reduced the promoting capabilities of the mutant

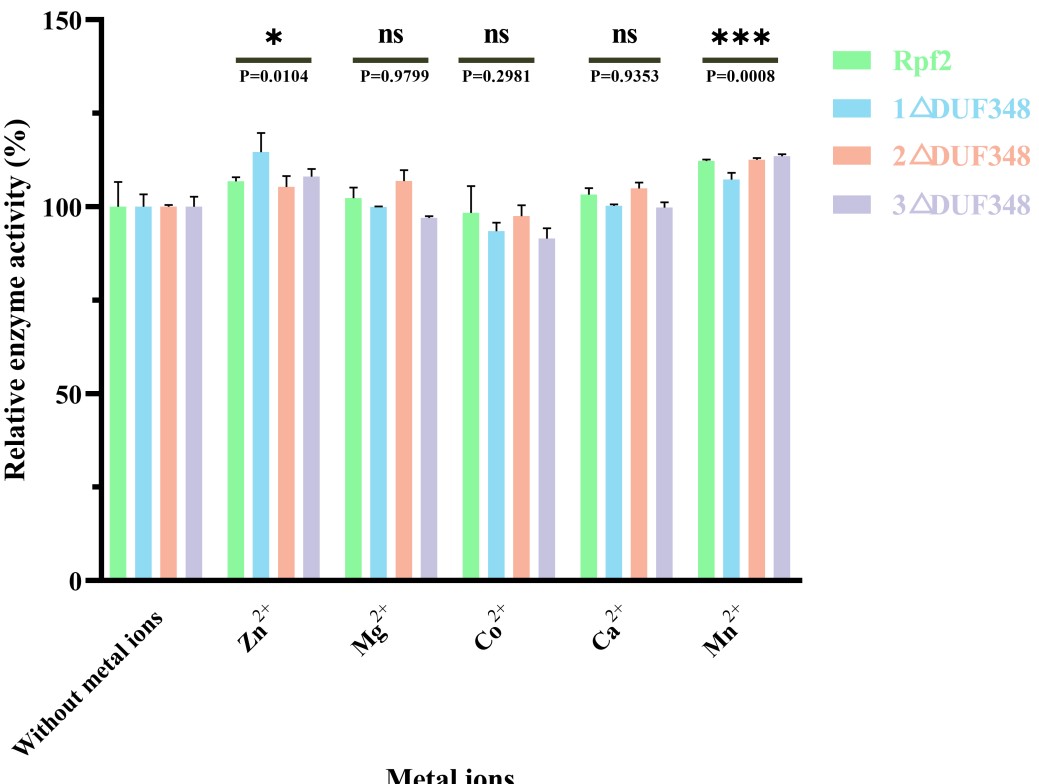

**Figure 6 Effects of the different metal ions on the muralytic activity of different Rpf2 proteins.** 1 △DUF348 Rpf2 represents Rpf2 with one missing DUF348 domain; 2 △DUF348 Rpf2 represents Rpf2 with two missing DUF348 domains; 3 △DUF348 Rpf2 represents Rpf2 with three missing DUF348 domains. The relative muralytic activity represents the ratio of the muralytic activity of each Rpf protein after the addition of metal ions to the muralytic activity of the Rpf protein without the addition of metal ions. ns, compared without metal ions means not significantly different.

proteins when compared with the wild-type Rpf2. Additionally, the promoting effect on the VBNC cells was also correlated with their muralytic activities.

## DISCUSSION

The Rpf proteins are crucial for cycling of the bacterial cell walls and formation of the cell membranes (*Ealand et al., 2018*; *Kwan & Qiao, 2023*). They function as a cytokine with muralytic activity, akin to small molecule cytokines. These proteins effectively promote the germination of dormant spores and affect the growth and sporulation of normal cells. The Rpf proteins show diversities in their consists of amino acids and molecular structures, *M. tuberculosis* and *S. coelicolor* contains five *rpf* genes (*rpf* A-E). The Rpf 2 of *R. erythropolis* KB1 has the same structure with Rpf B of *M. tuberculosis*, and possesses the largest molecular weight and the most complex structure (*Squeglia et al., 2013*). Similar to other Rpf proteins found in bacteria such as *M. luteus* (*Mukamolova et al., 2006*) and *M. tuberculosis* (*Mukamolova et al., 2002*), the Rpf protein in *R. erythropolis* KB1 also demonstrated growth-promoting and resuscitation effects on this strain. The

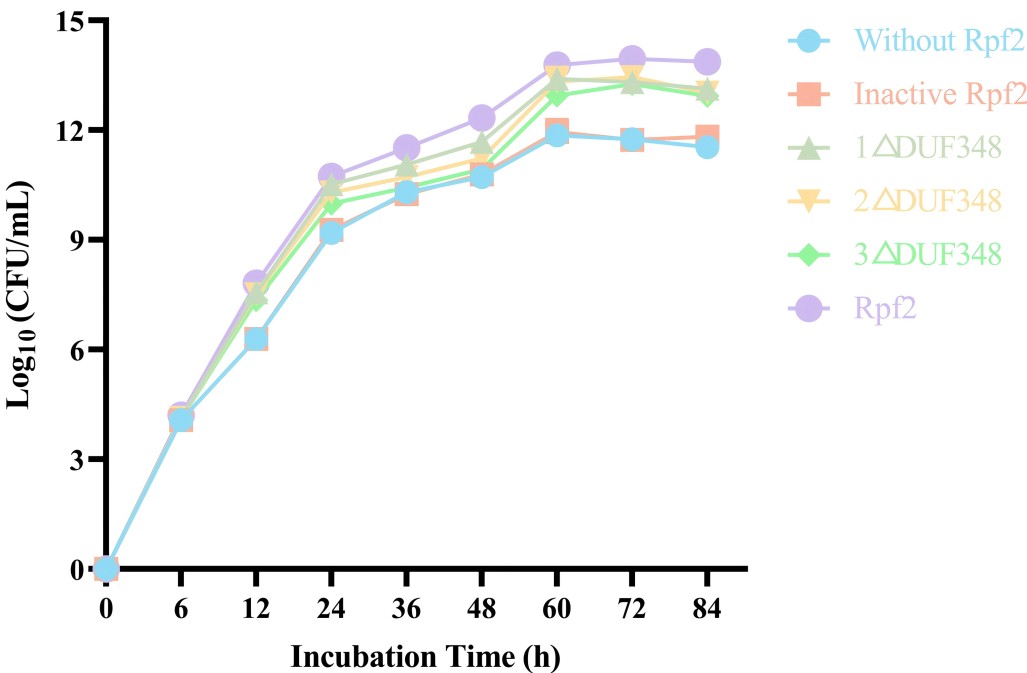

**Figure 7** Effects of the different purified recombinant Rpf2 on the growth of *R. erythropolis* KB1.
1 △DUF348 represents Rpf2 with one missing DUF348 domain; 2 △DUF348 represents Rpf2 with two missing DUF348 domains; 3 △DUF348 represents Rpf2 with three missing DUF348 domains.

Rpf2 contained the Rpf structure, G5 structure and three DUF348 domains. The DUF348 domain is conserved in some proteins and is essential for the enzyme activities (*Ruggiero et al., 2009*).

Here we elucidated the roles of the DUF348 structural domain on the muralytic activity and biological function of the Rpf2 in *R. erythropolis*. We found that the wild-type Rpf2 protein showed maximal muralytic activity. The progressive deletion of the DUF348 structural domains resulted in diminished enzyme activity and the growth-promoting effects. The activity of the Rpf2 mutant with three DUF348 domain deletion decreased to 42.56%. which suggested that the DUF348 domains important in enzymic activity and biological function. The structure analysis showed that deletion of the DUF348 domains resulted in a conformational change in the N-terminal. We suggested the following possibilities: First, the loss of structural domains altered the protein conformation, reducing its ability to bind to the cell wall. Second, the conformational changes affected the protein's interactions with substrates, or cofactors, or other proteins, and reduced the enzymatic efficiency. Previous research has proposed that the DUF348 domain forms an intriguing ubiquitin-like structure with the G5 domain, and suggested to enhance the interaction of the Rpf protein with peptidoglycan in the cell wall, thus facilitating effective cell wall interaction (*Ruggiero et al., 2016*). *Sexton et al. (2020)* researched the significance of the LysM and LytM domains in the enzyme activities of the RpfA and RpfD in *S. coelicolor*. They found that deletion of the LysM and LytM domains culminated in a mere 30.00%

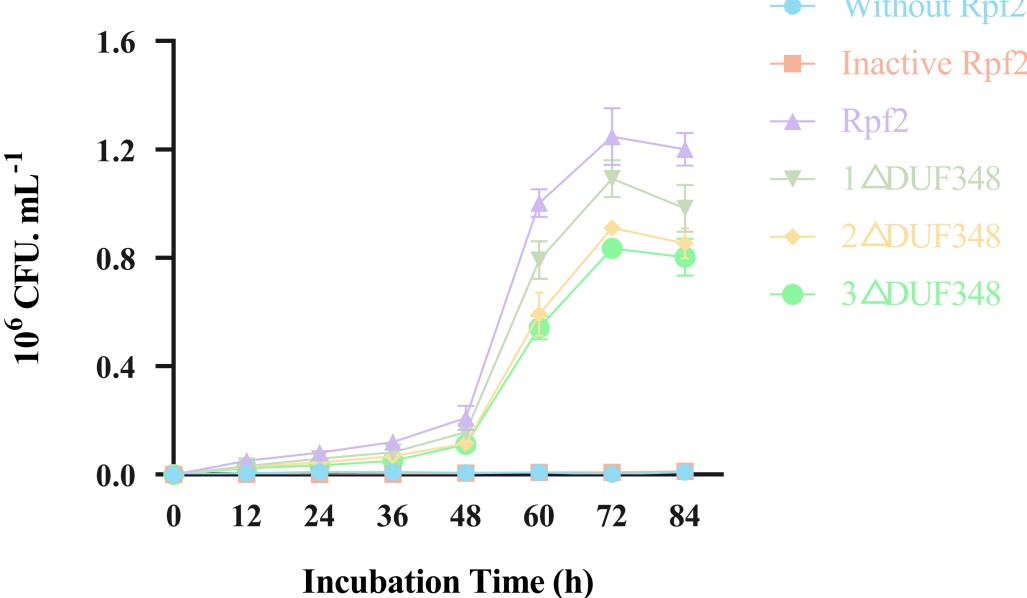

**Figure 8** **Effects of the different purified recombinant Rpf2 proteins on the resuscitation of VBNC *R. erythropolis* KB1.** 1 △DUF348 represents Rpf2 with one missing DUF348 domain; 2 △DUF348 represents Rpf2 with two missing DUF348 domains; 3 △DUF348 represents Rpf2 with three missing DUF348 domains.

of its original enzyme activity in RpfD, but deletion of the LysM domain in RpfA led to a reduction in the enzyme activity to approximately 70.00% (*Sexton et al., 2020*).

We also found that deletion of the DUF348 domains increased the susceptible to high temperature of the Rpf2 protein. The muralytic activity was more sensitive to high temperature, which also suggested that the DUF348 domains played important roles in maintaining the structural and functional integrity of the Rpf2 protein. $Zn^{2+}$ and $Mn^{2+}$ enhanced the muralytic activity of Rpf proteins. It is likely that $Zn^{2+}$ and $Mn^{2+}$ interacts with crucial residues within the enzyme's active site, thereby stabilizing the enzyme structure and optimizing substrate interactions and catalytic efficiency. $Mn^{2+}$ could enhance muralytic activity by interacting with certain amino acid residues, affecting its three-dimensional structure and catalytic sites (*Permyakov, 2021*). They might also play roles in substrate binding or modify the affinity between the enzyme and its substrate (*Kochańczyk, Drozd & Kr̨zel, 2015*).

## CONCLUSION

The present study examined the DUF348 domain of the Rpf2 protein in *R. erythropolis* KB1, emphasizing its critical role in maintaining enzymatic activity and stability. The deletion of DUF348 significantly reduced enzymatic activity and growth-promoting effects, thereby highlighting its importance to Rpf2 function. The muralytic activity of the protein remained stable between 20 °C and 40 °C. However, at higher temperatures, the muralytic activity of the Rpf protein lacking the DUF348 domain decreased more than that of the wild-type Rpf

protein, indicating that the DUF348 domain plays a crucial role in thermal stability. These findings contribute to our understanding of Rpf proteins and their potential applications in microbial resuscitation and bioremediation.

### Funding
This work was supported by grants from the National Natural Science Foundation of China (42267017), the Natural Science Foundation of Gansu Province ( 24JRRA970), the "Innovation Star" project for outstanding graduate students in Gansu Province (2023CXZX-421) and the Hongliu Young Talents Program of Lanzhou University of Technology. The funders had no role in study design, data collection and analysis, decision to publish, or preparation of the manuscript.

### Grant Disclosures
The following grant information was disclosed by the authors:
National Natural Science Foundation of China: 42267017.
Natural Science Foundation of Gansu Province: 24JRRA970.
Innovation Star" project: 2023CXZX-421.
Lanzhou University of Technology.

### Competing Interests
The authors declare there are no competing interests.

### Author Contributions
- Jianhui Fu conceived and designed the experiments, performed the experiments, analyzed the data, prepared figures and/or tables, authored or reviewed drafts of the article, and approved the final draft.
- Jixiang Chen conceived and designed the experiments, analyzed the data, prepared figures and/or tables, authored or reviewed drafts of the article, and approved the final draft.
- Yonggang Wang conceived and designed the experiments, authored or reviewed drafts of the article, and approved the final draft.
- Dan Luo conceived and designed the experiments, performed the experiments, analyzed the data, prepared figures and/or tables, authored or reviewed drafts of the article, and approved the final draft.
- Tianfeng Wang analyzed the data, authored or reviewed drafts of the article, and approved the final draft.
- Qingfang Zhang analyzed the data, authored or reviewed drafts of the article, and approved the final draft.

### DNA Deposition
The following information was supplied regarding the deposition of DNA sequences:

The Rpf2 sequence is available at GenBank: QIP41832.1. The Rpf2 sequence and its deletion of the DUF348 structural domain variant sequence are available in the Supplementary Files.

## Data Availability

The protein sequences (Rpf2 and variants lacking varying numbers of DUF348 structural domains) and the associated raw data are available in the Supplementary Files.

## Supplemental Information

Supplemental information for this article can be found online at http://dx.doi.org/10.7717/peerj.18561#supplemental-information.

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
