# Peer review of "The DUF348 domains of resuscitation promoting factor 2 play important roles in the enzymatic and biological activities in Rhodococcus erythropolis KB1"

_PeerJ, doi:10.7717/peerj.18561_

## Round 0.1 · original submission · Major Revisions

Dear authors,

After carefully considering the reviewers' comments and reviewing your manuscript, some aspects require major revisions. The most important issue is that the experiments were not carried out by calculating the same molar concentration, which can cause some variations. Also, a complementary technique such as circular dichroism unambiguously shows that the recombinant proteins have the proper folding, and the effect is associated with the deletions made, not because the protein has folding issues. I kindly request a careful revision of the manuscript and please include a point-by-point rebuttal letter to the issues detected.

Thank you so much for considering PeerJ for your research and all the best for your research moving forward.
Best regards,
Bernardo

Reviewer 1 ·

Basic reporting

The following minor points need to be resolved:

a) In line 155, they claim to have kept their cells for 2,000 days at 4ÂșC, in agreement with Dan Luo in his reference from lines 358 to 360; however, Luo only mentions having kept his cells for more than 1.350 days. Is the figure correct? In any case, the figure is missing a comma.

b) In the work, they never defined the acronym VBNC, and in the cited work by Luo, they defined VBNC as a viable but nonculturable state (VBNC). Is this meaning correct for the acronym? It is necessary to clarify

Experimental design

No comment

Validity of the findings

This work describes the behavior of a protein involved in the exit of the VBNC and touches on a phenomenon previously described as "Persistence." In any case, it is a biological phenomenon that has been poorly studied and understood but is of the utmost significance. It places the present work in a field of great novelty and importance.

Additional comments

The work entitled: The DUF348 domains of resuscitation promoting factor 2 play important roles in the enzymatic and biological activities in Rhodococcus erythropolis KB1, makes an adequate description of the functionality of said domain using constructions with successive deletions of three copies of the domain in the Rpf2 gene expressed in plasmid in cells stressed by cold shock. They determine the stability of the protein at different temperatures and divalent ions. The authors conclude that DUF348 is a domain that stabilizes the protein's enzymatic activity at various temperatures.

The paper is well written, the experiments are precise, and their conclusions are appropriate to their results.

Reviewer 2 ·

Basic reporting

Overall, the research work is carried out with sufficient scientific rigor. The raw data are indeed shown, and the structure of the work is easy to follow.

However, I suggest the abstract should have more background on R. erythropolis.

Experimental design

The experiments designed and carried out are suitable for testing the work's hypothesis, and the results shown support the conclusion reached.
A point that could help in the discussion section could be the following experiments: perform the modeling with the ions and observe the possible structural change conferred by the presence of the metal ions.

In Fig. 2, I think you should increase the definition of the structures; although they are observed adequately, they could have a better resolution.


Fig. 3 . The marker lacks units. Put which units they are.

Validity of the findings

The results support the conclusion of the work, and the experiments have their respective controls that allow the hypothesis to be verified.
However, I consider the conclusion section too extensive. Much of what can be handled in the discussion is mentioned in this or the results section. This section should be concise.

Additional comments

Line 20. Define VBNC
Line 20. Full name of R. erythropolis
Line 42. Define VBNC
Line 102. Is it agitation?
Line 151. change to its equivalent: rpm --> x g
Line 172. Full name of microorganisms.
Line 271. Full name of the microorganism.
Line 282. Ibidem.

Reviewer 3 ·

Basic reporting

no comment

Experimental design

no comment

Validity of the findings

no comment

Additional comments

In the manuscript "The DUF348 domains of resuscitation promoting factor 2 play important roles in the enzymatic and biological activities in Rhodococcus erythropolis KB1" the authors report the purification of a protein that has muralitic activity that promotes the resuscitation of the R. erythropolis bacteria when it is in the viable non-cultivable state.

The authors mention that the RpF protein (The resuscitation promoting factor) has several unknown domains (DUF348) that could be important in the protein function

It would be desirable for the authors to clarify where is the catalytic domain of the protein?. Is the rpf domain the catalytic domain? What is the G5 domain? In figure 1, an SP motif appears at the N-terminal. What is this sequence? Is it a Signal Peptide sequence for export to the periplasm?

The authors should indicate and add to the diagram in figure 1 which domains were eliminated from the Rpf protein.

Why was this combination of DUF348 domains eliminated? Motif 1 (the one closest to the N-terminal) was eliminated, as were domains 1 and 2 and domains 1, 2 and 3. Why was this combination? And not each of the domains independently or the combination of domains 2 and 3 or 1 and 3?

The figure caption for figure 1 should be clearer.

175 The authors mention that: "The N-terminal conformation of the Rpf2 undergoes alterations in the absence of the DUF348 domain, suggesting that this domain may play a critical role in maintaining protein stability and enzymatic efficiency (Fig. 2)" However, figure 2 does not indicate where is the N-terminal domain?. Where is the catalytic domain? Where are the DUF348 domains? Where is the G5 domain? Alphafold3 was used for the prediction. It would be desirable to show the score for each of the predictions.

The resolution of Figures 1 and 2 must be improved since what is shown is of low resolution.

A fundamental aspect of the work are the assays shown in Figures 5, 6, 7 and 8. In Figure 3 the authors show the various versions of the RpF protein used in this study, so it can be observed that the proteins have different molecular weights. In fact, the difference between the wild protein RpF and the delta3DUF348 mutant is approximately between 12 to 15 KDa. But the authors in the supplementary data say that the molecular weight is 39 KDa for the wild protein RpF and 28.6 for the delta3 version, but in Figure 3 very different apparent molecular weights are observed. How do you explain this?

The authors carried out all the experimental tests at the same concentration (0.3 mg/mL), but since the proteins have different molecular weights, they are not actually using equivalent amounts of each protein, that is, they are using different amounts of moles of each protein. From my point of view, in all the tests the same amount of millimoles or micromoles of each protein should be used and not milligrams of protein. Therefore, the results are not comparable and the conclusions could be wrong. For example, the wild protein is using fewer moles than the delta3DUF348 mutant, so the differences observed may be due to the fact that different amounts of each protein were used.

Mention what type of staining was used for the gel shown in figure 3.

The colors used in figures 4 and 5 to distinguish the different versions of the proteins are very similar and confuse the reader.

Why were these divalent ions used to see the enzymatic activity of the RpF versions?

Why was the color code in Figure 6 changed to that shown in Figures 4 and 5?

It seems that cations are not necessary for activity, since all versions are active. Cations only stabilize the structure and optimize the interaction with the substrate? BUT ARE THEY NOT CATION DEPENDENT? Is this common with other enzymes of the rpF family?

When observing Figure 7, it seems that the changes are not statistically significant. It would be desirable to show statistically that the results are significant. WHY NOT QUANTIFY GROWTH BY COUNTING COLONIES OR CALCULATING THE DUPLICATING TIME?

The result shown in Figure 8 is interesting since it shows that the presence of the RpF protein is necessary for the resuscitation of R. erythropolis cells. should be shown to be statistically significant between different versions of the Rpf proteins

---

## Round 0.2 · accepted · Accept

Dear authors,

Thank you for submitting this interesting work. I am happy to say that the reviewers and I agreed that the manuscript is now suitable for publication.
Congratulations and good luck with your research moving forward.

All the best,
Bernardo

Reviewer 1 ·

Basic reporting

no comment

Experimental design

no comment

Validity of the findings

no comment

Additional comments

After reviewing the authors' responses to my comments and those of other reviewers, I consider that the questions have been adequately answered; therefore, I suggest that the work be accepted for publication.

Reviewer 2 ·

Basic reporting

N/A

Experimental design

N/A

Validity of the findings

N/A

Additional comments

N/A

Reviewer 3 ·

Basic reporting

no comment

Experimental design

no comment

Validity of the findings

no comment

Additional comments

The authors have made the suggested corrections.